# Training and inference of large language models using 8-bit floating point

**Sergio P. Perez**,* **Yan Zhang**,* **James Briggs**,* **Charlie Blake**,
**Josh Levy-Kramer,  Paul Balanca,  Carlo Luschi,  Stephen Barlow,  Andrew Fitzgibbon**
Graphcore, United Kingdom

## Abstract

FP8 formats are gaining popularity to boost the computational efficiency for training and inference of large deep learning models. Their main challenge is that a careful choice of scaling is needed to prevent degradation due to the reduced dynamic range compared to higher-precision formats. Although there exists ample literature about selecting such scalings for INT formats, this critical aspect has yet to be addressed for FP8. This paper presents a methodology to select the scalings for FP8 linear layers, based on dynamically updating per-tensor scales for the weights, gradients and activations. We apply this methodology to train and validate large language models of the type of GPT and Llama 2 using FP8, for model sizes ranging from 111M to 70B. To facilitate the understanding of the FP8 dynamics, our results are accompanied by plots of the per-tensor scale distribution for weights, activations and gradients during both training and inference.

## 1   Introduction

Reducing the number of bits used by numerical formats offers significant efficiency gains for the training and inference of deep learning models. Inference latency is typically bottlenecked by the memory and communication bandwidth of a system [Pope et al., 2023], model-size by the total available memory, and throughput and training-time are often limited by the rate at which operations can be executed. All of these factors are improved substantially if we are able to represent values using fewer bits, with costs typically scaling linearly in the number of bits per value.

These benefits motivated the adoption of 16-bit floating-point formats — FP16 [Micikevicius et al., 2017] and BF16 [Kalamkar et al., 2019] — over the FP32 format used to represent continuous values for early deep learning models. More recently, 8-bit floating-point (FP8) formats have been proposed alongside hardware with dedicated support for FP8 arithmetic [Noune et al., 2022, Micikevicius et al., 2022], offering further efficiency gains. The standardisation of the FP8 format is under active development by the IEEE working group P3109 [2023]. The reader can find an introduction of floating-point formats for deep learning in Appendix A, and a description of the different FP8 formats in Appendix B. In this work, we assume the formats of Noune et al. [2022] when referring to FP8, denoting as FP8 E4 the weight and activation format and as FP8 E5 the gradient format.

These initial studies indicate that FP8 inference and training (that is, mixed-precision with matrix multiplications in FP8) are indeed possible, but come with a range of associated difficulties. Removing mantissa bits from a format limits numerical accuracy, while removing exponent bits limits the range of values that can be represented. The latter problem poses a particular challenge to practitioners: how to ensure that the set of values generated when performing model training and inference is within

---

*Corresponding authors: `{sergiop, yanz, jamesbr}@graphcore.ai`

Workshop on Advancing Neural Network Training at 37th Conference on Neural Information Processing Systems (WANT@NeurIPS 2023).

```
class LinearFP8Training:                          def weight_update(
                                                   w: fp16, dw: fp16
 # FP formats: fp8e4, fp8e5, fp16, fp32, bf16     ) -> fp8e4, int:
 # fp16 tensors can be replaced by fp32 or bf16    w: fp16 = optimiser(w, dw)
 fp8e5_max: fp8e5 = 57344                          w_scale: int = compute_bias(w, fp8e4)
 fp8e4_max: fp8e4 = 240                            w: fp16 = scale(w, w_scale)
                                                   w8: fp8e4 = cast(w, fp8e4)
 def forward(                                       return w8, w_scale
  w8: fp8e4, w_scale: int, x: fp16
 ) -> fp16:                                        def compute_bias(
  x_scale: int = compute_bias(x, fp8e4)            tensor: fp16,
  x: fp16 = scale(x, x_scale)                      cast_to: Union[fp8e4, fp8e5],
  x8: fp8e4 = cast(x, fp8e4)                        margin: int = 3 # See Subsection 3.2
  y: fp16 = matmul(x8, w8.T)                       ) -> int:
  y: fp16 = unscale(y, x_scale + w_scale)          amax: fp16 = max(abs(tensor))
  return y                                          if cast_to == fp8e4:
                                                     return floor(log2(fp8e4_max/amax)) - margin
 def backward(                                      elif cast_to == fp8e5:
  dy: fp16, w8: fp8e4, w_scale: int,                return floor(log2(fp8e5_max/amax)) - margin
  x8: fp8e4, x_scale: int
 ) -> fp16, fp16:                                  def scale(
  dy_scale: int = compute_bias(dy, fp8e5)          v: fp16, v_scale: int
  dy: fp16 = scale(dy, dy_scale)                   ) -> fp16:
  dy8: fp8e5 = cast(dy, fp8e5)                      return v * 2**v_scale
  dx: fp16 = matmul(dy8, w8.T)
  dx: fp16 = unscale(dx, dy_scale + w_scale)      def unscale(
  dw: fp16 = matmul(dy8, x8.T)                     v: fp16, v_scale: int
  dw: fp16 = unscale(dw, dy_scale + x_scale)      ) -> fp16:
  return dx, dw                                     return v * 2**(-v_scale)
```

Figure 1: Training phase of a linear layer quantised to FP8. The forward and backward pass illustrate how the scaling biases are computed and applied to the weights, activations and gradients.

```
from LinearFP8Training import compute_bias, scale  unscale
# fp16 tensors can be replaced by fp32 or bf16

class LinearFP8Inference:                          def forward(
                                                    w8: fp8e4, w_scale: int, x: fp16
 def post_training_quantisation(                   ) -> fp16:
  w: fp16 # Checkpoint                               x_scale: int = compute_bias(x, fp8e4)
 ) -> fp8e4, int:                                   x: fp16 = scale(x, x_scale)
  w_scale: int = compute_bias(w, fp8e4)            x8: fp8e4 = cast(x, fp8e4)
  w: fp16 = scale(w, w_scale)                      y: fp16 = matmul(x8, w8.T)
  w8: fp8e4 = cast(w, fp8e4)                       y: fp16 = unscale(y, x_scale + w_scale)
  return w8, w_scale                                return y
```

Figure 2: Inference phase of a linear layer quantised to FP8. Post-training quantisation is applied to a checkpoint. Scaling biases are computed and applied to the weights and activations.

the set of representable values. Overflowing or underflowing this range can rapidly degrade model accuracy.

To combat this for FP16 training, the standard approach is to globally shift gradients by a single *loss scale* [Micikevicius et al., 2017, Noune et al., 2022, Perez, 2022], though this is not always sufficient [Zhang et al., 2022, Scao et al., 2022]. For inference, a popular technique is quantisation to the 8-bit *integer* format (INT8). Previous generations of AI hardware have offered accelerated arithmetic for INT8 but not FP8, limiting FP8 uptake despite its potential as a more broadly-applicable 8-bit format in the context of machine learning (see Appendix C for further discussion). More complex group-quantisation schemes have also been proposed for inference which enable some values to be stored in fewer than 8 bits [Dettmers and Zettlemoyer, 2022]. However, this introduces additional complexity and compute must still be done in higher-precision.

To address the issue of substantially reduced range for FP8 formats, it has been proposed to rely on the exponent bias associated with FP8 tensors. The exponent bias is part of the definition of every floating-point format. By adding or subtracting an integer to the exponent bias, one can effectively shift the representable range on a per-tensor basis, giving more granular scaling than standard *loss scaling* and applying to both forward and backward passes. This integer, denoted as *scaling bias*, is supplied by the user and can be supported either in software or directly in hardware.

The process by which these scales are determined and how they are practically applied is essential to leveraging the benefits of FP8 for training and inference. Existing FP8 literature has not covered this topic extensively, leaving users reliant on scaling decisions taken in software implementations that may not be clearly justified [Nvidia, 2022b]. We seek to support this important design aspect through the following contributions:

1. We present a methodology to select the per-tensor scaling biases in the linear layers present in large language models of the type of GPT [Brown et al., 2020] and Llama [Touvron et al., 2023]. Such methodology is illustrated in Figure 1 for the training phase and in Figure 2 for the inference phase. These specific details are useful for practitioners aiming to leverage FP8 and have been missing from the FP8 literature, which has either employed sweeps of values [Noune et al., 2022] or not specified how the scaling biases are computed [Micikevicius et al., 2022].

2. We showcase how our FP8 methodology leads to convergence of GPT and Llama models from 111M to 70B parameters, for both inference and training.

3. For inference, we detail how our methodology can be employed as post-training quantisation to cast a high-precision checkpoint to FP8 and perform inference without degradation.

4. For training, we prove that our methodology is able to dynamically update the per-tensor scaling biases and prevent degradation using FP8 in large language models. We provide plots of how the scaling biases evolve and extract insights from them.

## 2 The linear layer adapted to FP8

Performing the matrix multiplication operation in FP8 requires the use of *scalings* to prevent underflow and overflow. By *scalings* we mean factors that, when multiplied times a tensor, result in a scaled tensor representable in the FP8 dynamic range. Without such scale, the tensor underflows or overflows. Such scalings are needed for the matrix multiplications found in both the forward pass (to compute the activations) and in the backward pass (to compute weight and activation gradients). Using scalings for lower precision is not new and has been a popular strategy for FP16 training, with the loss scaling method [Noune et al., 2022, Perez, 2022, Micikevicius et al., 2017] consisting of multiplying the loss function with a constant to prevent underflow of the gradients. Although loss scaling works fine for reasonably sized FP16 models, as the number of parameters increases the limited range of FP16 becomes an issue. Models of more than 100 billion parameters like Bloom [Scao et al., 2022] or OPT [Zhang et al., 2022] struggled to find a stable loss scaling for FP16 and ended up employing BF16. Consequently, it's uncertain whether even for FP16 it is enough to have a common scaling for all the gradients. The same question has been explored for FP8: it is not clear whether one scaling is enough [Noune et al., 2022] or a per-tensor scaling is needed [Micikevicius et al., 2022]. In addition, for FP8 E4 weights and activations also need scalings due to the reduced dynamic range compared to FP16.

Figure 3 illustrates how the scalings are implemented for the forward pass of a FP8 linear layer. Firstly, focusing on the full FP16 precision, Figure 3a displays both weights and activations in FP16 and no scaling is needed before the matrix multiplication, whose accumulation can be performed in FP16 too. This scenario is identical for other formats like FP32 or BF16. In comparison, Figure 3b shows how different scaling and casting blocks are needed to leverage FP8 matrix multiplication in mixed precision. The inputs are FP8 but the output is FP16: this dichotomy comes from the need of accumulating the partial results of the FP8 operations in FP16 to prevent overflows. Since the accumulation is in FP16, hardware providers [Graphcore, 2022b, Nvidia, 2022a] output the internal FP16 result and let the user decide whether to cast back down to FP8.

**Weights**    For training and inference, the linear layer needs to be modified to include a cast to FP8 E4 from a higher-precision format like FP16. In training, this cast is necessary after every weight

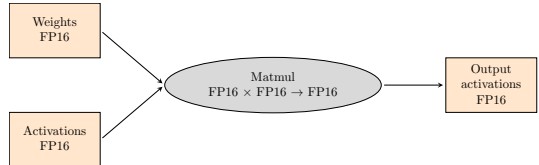

(a) Forward pass for FP16 inference.

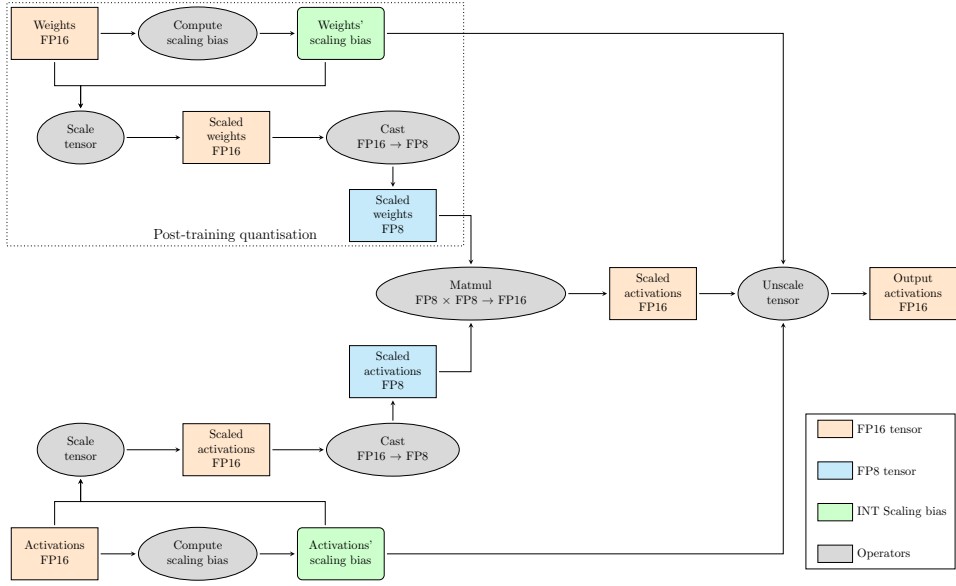

(b) Forward pass for FP8 inference.

Figure 3: Comparison of the forward pass for a FP16 vs FP8 linear layer.

update, which takes place in a higher-precision format like FP16 or FP32. In inference, if the weights are stored in FP8 then no cast is needed. Conversely, if the weights are in a higher-precision format like FP16, BF16 or FP32, a cast to FP8 E4 is needed just once before using those weights in the matrix multiplication. For both cases, before the cast to FP8 E4, a scaling of the weights is needed to prevent underflow or overflow when performing such cast. The scaling shifts the weight distribution and makes it overlap as much as possible with the dynamic range of FP8 E4. The optimal scalings may change during training so there's the need to recompute the scaling again after a certain number of steps. During inference, the scalings don't change since the weights are not updated.

**Activations**  Due to the matrix multiplication accumulation being done in higher precision, it is necessary to cast back to FP8 E4 before the next matrix multiplication. When casting to FP8 E4, we need a scaling factor to minimize the underflow/overflow since the dynamic range of FP8 E4 is narrower compared to higher-precision formats like FP16. After the matrix multiplication is performed, the output activations are unscaled taking into account the scaling factors computed for the weights and activations before the matrix multiplication.

## 2.1 Applying a scaling bias before casting to FP8

Casting the weights and activations from FP16 to FP8 E4 results in a narrower dynamic range that may lead to underflow or overflow. To prevent it, we introduce per-tensor scalings that shift the FP16 distributions before casting to FP8 E4. The type of scaling employed in this work is a *scaling bias*. Starting from the floating point representation defined in Equation 4, we add an integer scaling bias $b_{\text{scale}}$ to the exponent such that

$$\text{scaled exponent} = b_{\text{exp}} - \text{bias} + b_{\text{scale}}, \tag{1}$$

which is equivalent to multiplying the FP16 number times $2^{b_{\text{scale}}}$. Both the weights and activations in Figure 3b require a scaling bias before being cast from FP16 to FP8 E4. Let's denote as $b_{\text{w,scale}}$ and $b_{\text{x,scale}}$ the scaling biases for the weights and activations, respectively. Then, once the matrix multiplication is performed in a higher-precision accumulation like FP16, the resulting activations need to be unscaled by applying a scaling bias equal to $-(b_{\text{w,scale}} + b_{\text{x,scale}})$:

$$\text{unscaled exponent} = b_{\text{exp}} - \text{bias} - (b_{\text{w,scale}} + b_{\text{x,scale}}). \tag{2}$$

We refer the reader to the `scale` and `unscale` functions in Figure 1, which are employed in the code for the training and inference phases in Figures 1 and 2.

## 2.2 FP8 for gradients during training

The backward pass for the linear layer contains two matrix multiplications: one to compute the weight gradients and another for the input activation gradients. Both matrix multiplications can be accelerated with FP8. The process is similar to the matrix multiplication in the forward pass: the inputs of the matrix multiplication need to be scaled and then cast to FP8 before being passed to the matrix multiplication. Subsequently, the matrix multiplication output (i.e the weight gradients or activation gradients) are unscaled taking into account the scales of the FP8 matrix multiplication inputs. It's important to recall that the FP8 type is different for weights and activations versus gradients: whereas the weights and activations are cast to FP8 E4, the gradients need to be cast to FP8 E5 to preserve a wider dynamic range (see Appendix B for the differences between the two formats). We refer the reader to the pseudocode in Figure 1 for details about the `backward` pass in FP8.

## 2.3 Choosing the appropriate scaling bias

There are various methods to quantise from a higher-precision format into a lower one. Some popular approaches to cast from a floating point format like FP32 into a fixed-point format like INT8 consist of mapping the largest absolute value to $\pm 127$, which is the maximum representable integer in INT8. This ensures that the outliers fit within the dynamic range of INT8, but may underutilise the dynamic range if the outliers are much larger than the other values. Other approaches consider a percentile or the full distribution of values and compute the mean square error or KL divergence to minimise the information loss between the higher-precision distribution and the quantised one [Settle et al., 2018].

In this work we propose a methodology based on setting dynamic per-tensor scalings, computed via the absolute maximum approach. Our strategy has similarities to the Nvidia [2022b] library; however some of the fine-grained details and justifications of this implementation are not made explicit. We hope that by opening up our methodology and testing it in the experiments in Section 4, other FP8 researchers can build on top of it.

Our methodology depends on the maximum representable number of the FP8 format, which is different for the FP8 E4 and FP8 E5 formats (see Appendix B). Denoting that maximum as $\text{max}_{\text{num}}$, the calculation of the scaling bias per tensor follows

$$
\begin{aligned}
\text{amax} &= \max\left(|\text{tensor}|\right), \\
\text{scaling\_bias} &= \text{floor}\left(\log_2\left(\text{max}_{\text{num}}/\text{amax}\right)\right),
\end{aligned}
\tag{3}
$$

where $\text{floor}(a)$ returns the largest integer not greater than a. The function `compute_bias` in Figure 1 translates this algorithm into code. For training (see Figure 1), three scaling biases are computed in each linear layer, corresponding to the weights, input activations and output activation gradients. For inference(see Figure 2), only the weight and input activation need scaling biases.

## 2.4 Loss scaling in addition to scaling bias when accumulating in FP16

Loss scaling is a popular technique to enable FP16 training [Noune et al., 2022, Perez, 2022, Micikevicius et al., 2017]. Loss scaling is necessary in FP16 because the gradients underflow due to the narrower dynamic range of FP16, compared to other formats like FP32 or BF16. The reason because of which loss scaling is also relevant for FP8 quantisation is related to the higher-precision accumulation of the FP8 matrix multiplication. Such accumulation is usually performed in FP16, BF16 or FP32 [Graphcore, 2022b, Nvidia, 2022a]. If it was done in FP8, it wouldn't work due to the limited dynamic range for FP8 E4 or the lack of precision in FP8 E5. As a result, the linear layer quantisation to FP8 described in this section is actually mixed-precision quantisation.

When the accumulation is performed in BF16 or FP32, loss scaling is not necessary and just the scaling biases explained in Subsection 2.3 are enough to prevent underflow or overflow after casting to FP8. However, when the accumulation is performed in FP16, loss scaling is needed to better represent the gradients after they are output by the FP8 matrix multiplication and unscaled. The method to tune the loss scaling for mixed FP8-FP16 training is identical to the full FP16 training. There are several approaches in the literature: run a sweep of loss scalings [Micikevicius et al., 2017], inspect the gradient histogram to adapt the loss scaling during training [Perez, 2022], back off to skip weight updates when an overflow is produced, or scale the loss such that its mean plus standard deviation times a constant equals $log_2$ of the maximum representable value in FP16 [Kuchaiev et al., 2018]. We refer the reader to section 4 of [Noune et al., 2022] for an analysis about how these loss scaling methods impact mixed FP8-FP16 training. In our experiments in Section 4, we use a constant loss scaling, using the same values for the full FP16 training and mixed FP8-FP16 training.

# 3   Details to perform training and inference in FP8

We follow two different strategies to compute the scaling bias for training and inference:

- FP8-AMAX: this is the absolute maximum method detailed in Section 2.3 and in the `compute_bias` function of Figure 1. The calculation takes place per linear layer for every micro batch and every data or tensor replica, following the diagram in Figure 3b.

- FP8-CSCALE: a simpler strategy based on having the same scaling bias for all weights, activations and gradients. The scaling bias remains constant throughout the training and inference. We run sweeps of scaling bias values to find the ones that don't degrade accuracy.

Even though in this paper we focus on the numerical differences, it is worth pointing out that the relative throughput and memory cost of FP8-AMAX versus FP8-CSCALE depends on the hardware employed. When using FP8-AMAX in hardware with limited SRAM, FP16 tensors in L2-cache incur the overhead of a second round-trip to memory: the first to calculate the tensor's absolute max, and the second to apply the scaling. This cost could cancel out the speedup from the FP8 matmuls. A remedy could be to rely on the past history of absolute max instead of using the just-in-time absolute max Nvidia [2022b]. On the contrary, hardware with enough SRAM can calculate the scaling biases just-in-time and perform FP8 as detailed in this work.

## 3.1   Inference with FP8

When performing inference, the weights come from a checkpoint that is either in a higher-precision format like FP16, BF16 or FP32, or directly in FP8 E4. In the former case, quantising the weights to FP8 is simpler compared to fixed-point representations like INT8, which may need quantisation-aware training (QAT) in addition to post-training quantisation (PTQ) [van Baalen et al., 2023]. For FP8, it is enough to employ PTQ consisting of applying a scaling bias to each tensor and subsequently casting to FP8, as described in Section 2.3. The scaling bias calculation for the weights is performed only once when loading the checkpoint (see Figure 2). In the latter case, when the checkpoint comes from training in FP8, the weights can be used directly without any quantisation.

## 3.2   Training with FP8

For pre-training or fine-tuning, we need different FP8 formats for the weights/activations and gradients (see Appendix B and Noune et al. [2022]). For both formats, we compute the scaling bias following either the FP8-AMAX or the FP8-CSCALE, as stated in each of the experiments in Section 4. We perform the weight update in FP16 and keep master weights in FP16. The calculation of the scaling bias for the weights and the weight cast to FP8 E4 takes place just after the weight update. When accumulating in FP16, there's a risk of overflowing when performing the two matrix multiplications of the backward pass, which have inputs FP8 E4 and FP8 E5: this is due to the fact that FP8 E5 and FP16 have a similar dynamic range (see Table 7), and when employing FP8-AMAX the resulting FP8 E5 input to the matmul gets values closer to the maximum representable number in FP16. Consequently, we set a *margin* to reduce the scaling bias resulting from FP8-AMAX method. Empirically we observe that a value of 3 is enough to prevent overflow. The optimal value for this margin is related to the square root of the batch size [Blake et al., 2023, Yang et al., 2021], which in our fine-tuning

Table 1: Hierarchy of GPT and Llama 2 model sizes used in the training and validation experiments.

| Parameters | $d_{\text{model}}$ | $n_{\text{layers}}$ | $n_{\text{heads}}$ | $d_{\text{head}}$ | $d_{\text{ffn}}$ |
|---|---|---|---|---|---|
| GPT 111M | 768 | 10 | 12 | 64 | 3072 |
| GPT 590M | 1536 | 18 | 12 | 128 | 6144 |
| GPT 1.3B | 2048 | 24 | 16 | 128 | 8192 |
| GPT 6.7B | 4096 | 32 | 32 | 128 | 16384 |
| GPT 13B | 5120 | 40 | 40 | 128 | 20480 |
| Llama 2 7B | 4096 | 32 | 32 | 128 | 11008 |
| Llama 2 70B | 8192 | 80 | 64 | 128 | 28672 |

experiments is 512 (see Appendix H). This results in a optimal margin of $log_2(\sqrt{512}) = 4.5$, which is close to our empirical value of 3.

## 4  Experiments

### 4.1  Model architecture used for the experiments

We employ two varieties of language transformer decoder models in our experiments. The first one is a GPT-3-like architecture [Brown et al., 2020] with the sole difference of using dense attention in all decoder blocks, instead of dense and sparse-banded attention. For this model we test five different model sizes (see Table 1). In our fine-tuning experiments, we employ the pre-trained checkpoints provided by Dey et al. [2023]. In our inference experiments, we start from an already fine-tuned checkpoint in FP16 for each specific task. We focus on three GLUE tasks [Wang et al., 2018]: the inference task MNLI, the single-sentence task SST-2 and the similarity and paraphrase task QQP.

The second variety of decoder language model is the Llama 2 model detailed in Touvron et al. [2023]. The main changes with respect to the GPT-3-like architecture are the pre-normalization using RMSNorm, SwiGLU as activation function and rotary positional embeddings. In addition, the 70-billion-parameter version employs grouped-query attention. We employ the open-source checkpoints from the pre-trained models that are not fine-tuned for dialogue use cases. The details of the 2 sizes tested in our experiments are shown in Table 1. We focus on six benchmarks included in Touvron et al. [2023]: MMLU, HellaSwag, ARC-e, ARC-c, PIQA and WinoGrande.

For both architectures, we quantise to FP8 the linear layers in all the decoder layers. Details about such linear layers are shown in Appendix E. Figure 4 displays the main components of the GPT and Llama decoder layers and indicates the ones quantised to FP8. Further details about hyperparameters and hardware to run the experiments are contained in Appendix H.

### 4.2  FP8 inference for the GPT model

We compare the validation results using the FP8-AMAX and FP8-CSCALE methods versus the FP16 benchmark, for a GPT model with sizes from 111M to 13B. The results are displayed in Table 2. With both approaches we manage to match the FP16 validation accuracy for all sizes.

For the FP8-CSCALE method, we run sweeps of scaling biases. Not all the scaling biases reach the FP16 accuracy, and in Table 2 we report the average accuracy obtained with only the values that reach a final accuracy greater than 99.5% of the FP16 value. The interval containing the convergent values is displayed in Table 3. For the scaling bias values outside the intervals in Table 3, the validation accuracy degrades significantly. In Figure 5 in Appendix F we show a comparison of the accuracy obtained with each of the scaling bias in the sweep, for the MNLI task. As soon as the chosen scaling bias is not within the interval, it quickly degrades. On average we observe that the interval of convergent scaling bias values contains five integers centred around zero.

For the FP8-AMAX method, there's a different scaling bias for each weight and activation tensor. To understand how the different scaling biases vary depending on the decoder layer and type of linear layer, we plot their distributions in Figure 6 for the 111M, 1.3B and 6.7B parameter models. The

Table 2: Inference results: validation accuracy comparing FP16 with FP8-AMAX and FP8-CSCALE, for the different GPT model sizes.

| Model | Quantisation | MNLI | QQP | SST-2 |
|---|---|---|---|---|
| 111M | FP16 | 72.61 | 85.76 | 84.26 |
| | FP8-AMAX | 72.39 | 85.78 | 84.38 |
| | FP8-CSCALE | 72.49 | 85.73 | 84.59 |
| 590M | FP16 | 78.59 | 88.40 | 90.63 |
| | FP8-AMAX | 78.44 | 88.37 | 90.63 |
| | FP8-CSCALE | 78.56 | 88.40 | 90.54 |
| 1.3B | FP16 | 82.82 | 89.43 | 91.55 |
| | FP8-AMAX | 82.68 | 89.42 | 91.44 |
| | FP8-CSCALE | 82.72 | 89.36 | 91.42 |
| 6.7B | FP16 | 87.17 | 91.19 | 94.50 |
| | FP8-AMAX | 87.15 | 91.22 | 94.38 |
| | FP8-CSCALE | 87.18 | 91.18 | 94.48 |
| 13B | FP16 | 88.26 | 91.22 | 94.61 |
| | FP8-AMAX | 88.27 | 91.21 | 94.61 |
| | FP8-CSCALE | 88.26 | 91.20 | 94.50 |

Table 3: Inference results with FP8-CSCALE: range of the scaling bias that reaches a validation accuracy greater than 99.5% of the FP16 value, when performing FP8 validation with FP8-CSCALE. Both weights and activations in all decoder layers share the same scaling bias.

| Model | MNLI | QQP | SST-2 |
|---|---|---|---|
| 111M | [-3, 2] | [-4, 2] | [-4, 2] |
| 590M | [-3, 2] | [-4, 2] | [-1, 2] |
| 1.3B | [-3, 3] | [-4, 2] | [-3, 2] |
| 6.7B | [-3, 2] | [-3, 2] | [-3, 2] |
| 13B | [-3, 2] | [-4, 2] | [-4, 2] |

Table 4: Inference results of Llama 2. For the evaluation we follow Touvron et al. [2023], performing 5-shot evaluation for MMLU and 0-shot evaluation for HellaSwag, ARC-e, ARC-c, PIQA and WinoGrande. For WinoGrande we report the accuracy and for MMLU, HellaSwag, ARC-e, ARC-c and PIQA the normalized accuracy, which takes into account the lenght of each possible answer.

| Model | Quantisation | MMLU | HellaSwag | ARC-e | ARC-c | PIQA | WinoGrande |
|---|---|---|---|---|---|---|---|
| 7B | Llama 2 paper | 45.3 | 77.2 | 75.2 | 45.9 | 78.8 | 69.2 |
| | FP16 | 46.6 | 76.0 | 74.6 | 46.3 | 79.1 | 69.1 |
| | FP8-AMAX | 46.3 | 75.8 | 74.5 | 45.7 | 78.7 | 69.1 |
| 70B | Llama 2 paper | 68.9 | 85.3 | 80.2 | 57.4 | 82.8 | 80.2 |
| | FP16 | 69.6 | 83.8 | 81.1 | 57.3 | 82.8 | 78.0 |
| | FP8-AMAX | 69.3 | 83.8 | 80.9 | 57.7 | 82.6 | 78.5 |

reader can find details about how Figure 6 is produced in Appendix G, together with some insights about the scaling bias distribution.

## 4.3 FP8 few-shot inference for the Llama 2 model

We run six of the evaluation benchmarks in Touvron et al. [2023] with both FP16 and FP8-AMAX, for the model sizes of 7B and 70B parameters. For the benchmarks we employ Eleuther's Evaluation Harness Library [Gao et al., 2021]. The results are displayed in Table 4. We find that the FP16 and FP8-AMAX quantisations give comparable results. For some benchmarks like HellaSwag there is some difference with respect to the result published in Touvron et al. [2023], which we attribute to the fact that the authors employ an internal evaluation library different from Gao et al. [2021]. We checked this by comparing the harness' benchmark results in FP32 running on CPU to those obtained with FP16 and confirmed that the metrics obtained are identical.

## 4.4 Is FP8-CSCALE enough to train in FP8?

Running sweeps of loss scaling values is a common practice to train models in FP16. As the size of the model increases, one typically needs to increase the loss scaling value. Even though there exists

Table 5: Fine-tuning results: validation accuracy after fine-tuning in FP16 and FP8-AMAX for 3 epochs.

| Model | Quantisation | MNLI | QQP | SST-2 |
|-------|--------------|------|------|-------|
| 111M | FP16 | 72.61 | 85.32 | 85.07 |
| | FP8-AMAX | 72.50 | 85.84 | 85.57 |
| 590M | FP16 | 78.59 | 88.25 | 89.27 |
| | FP8-AMAX | 79.12 | 88.31 | 89.00 |
| 1.3B | FP16 | 82.82 | 89.32 | 91.36 |
| | FP8-AMAX | 82.58 | 89.32 | 91.28 |
| 6.7B | FP16 | 87.17 | 91.19 | 94.53 |
| | FP8-AMAX | 87.26 | 91.06 | 94.84 |
| 13B | FP16 | 88.26 | 91.22 | 94.61 |
| | FP8-AMAX | 88.28 | 91.53 | 94.50 |

Table 6: Fine-tuning results with FP8-CSCALE: range of the scaling bias that reaches a validation accuracy greater than 99.5% of the FP16 value, when performing FP8 fine-tuning with FP8-CSCALE. Weights, activations and gradients in all decoder layers share the same scaling bias.

| Model | MNLI |
|-------|--------|
| 111M | [-3, 2] |
| 590M | [-2, 2] |
| 1.3B | [-2, 1] |
| 6.7B | [-1, 1] |
| 13B | [-1, 0] |

more sophisticated approaches to update the loss scaling during training [Perez, 2022, Kuchaiev et al., 2018], practitioners still run sweeps of loss scaling values until finding the one that converges.

Inspired by this practice, we aim to understand if the FP8-CSCALE approach is able to converge to the required accuracy. For that we run sweeps of values and let the fine-tuning for the MNLI task complete three epochs for the smaller models up to 1.3B and 1 epoch for the 6.7B and 13B. Then we check if the validation accuracy matches the reference FP16 fine-tuning.

Our results are summarised in Table 6. We are able to converge to a validation accuracy of at least 99.5% of the FP16 reference for all the model sizes, but as the size increases the range of converging scaling biases gets reduced. For the larger model sizes of 6.7B and 13B, we observe that convergence is not always guaranteed even within the intervals in Table 6: for example, a different seed can lead to divergence. These results suggest that FP8-AMAX is a more robust strategy when fine-tuning in FP8 compared to FP8-CSCALE, even though convergence with FP8-CSCALE may be possible.

### 4.5 FP8 fine-tuning results for the GPT model

After testing FP8-CSCALE, we employ the FP8-AMAX method to fine-tune the GPT models for the sizes from 111M to 13B. With FP8-AMAX we are able to converge fine for all the sizes tested and the three different GLUE tasks of MNLI, QQP and SST-2, when compared to the validation accuracy reached in FP16. The results are displayed in Table 5. The loss function evolution also converges similarly when comparing FP8-AMAX and FP16. The loss function plots for the MNLI task are shown in Figure 10 within the Appendix J.

In Appendix I we provide plots and analysis about how the scaling biases evolve as the fine-tuning progresses, for the model sizes of 111M, 1.3B and 6.7B. Inspecting the per-tensor scalings resulting from FP8-AMAX is helpful to elucidate why the FP8-CSCALE strategy in Subsection 4.4 is not robust for large models. It also gives insights about the update frequency needed if one is interested in saving some of the extra computations needed to update the scaling bias with FP8-AMAX.

## 5 Conclusion

We provide the technical details for practitioners interested in leveraging FP8 quantisation to effectively employ it for inference and training. We show that our methodology is able to adapt the scaling biases to prevent underflow or overflow from the FP8 format and match the reference results obtained in higher precision, for large language models like GPT and Llama up to 70B parameters.

In this work we have focused on quantising the linear layers to FP8, but there are other layers ubiquitous in most transformer architectures that may benefit from FP8 quantisation, like the dot-product attention. We'll explore those in future works, as well as the application of FP8 in other models that don't belong to the transformer family of architectures, such as graph neural networks or computer vision models based on convolutional layers.

## Acknowledgments and Disclosure of Funding

We would like to thank the following engineers at Graphcore for their contributions to the paper at the various stages of its development: Matthew Haddock, Shiraz Butt, Artemiy Bulavin, Mark Kattenbelt, Godfrey Da Costa, Jake Hall, Tim Poole, Douglas Orr, Graham Horn, Ian Hales, Sylvain Viguier, Anjlee Gopiani, Arsalan Uddin and Manuele Sigona.

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

# A Floating point formats in deep learning

The floating point format convention is defined by the IEEE 754 standard [IEEE, 2019]. There are several definitions of floating point formats, which differ in the total number of bits to represent them and how the bits are distributed between the exponent bits ($E$) and the mantissa bits ($M$). The general formulation for a floating point value is

$$\text{value} = (-1)^{\text{sign}} \times 2^{\text{exponent}} \times \text{mantissa}, \tag{4}$$

where $\text{sign} \in \{0, 1\}$ is the sign bit, $\text{exponent} = b_{exp} - \text{bias}$ with $b_{exp}$ being a bit-string with $E$ bits and $\text{bias} = 2^{E-1} - 1$, and $\text{mantissa} = 1 + \sum_{i=0}^{M} d_i 2^{-i}$ with $d_i \in \{0, 1\}$. These also exist some special values represented by bit-strings that don't follow the above interpretation. These are the infinities, NaN (not-a-number) and subnormal numbers to represent even smaller absolute values.

The common floating point formats used in deep learning are compared in Table 7. The number of total bits in each format is equal to the sum of the sign bit (always 1), $E$ and $M$. Whereas many applications in scientific computing require at least double precision (i.e. FP64) to preserve accuracy, in deep learning it is enough to use single precision (i.e. FP32). However, formats employing fewer bits have received much attention due to the prospect of increasing the number of operations per cycle, reducing the memory to store weights, activations or gradients and alleviating bandwidth constraints. At the same time, a lower number of bits can impact numerical accuracy and the dynamic range.

We denote by "low precision" all the numerical formats with less than 32 bits. The main reason low-precision formats can degrade performance is because of their reduced dynamic range. Whereas FP32 can represent normal values approximately in the interval $[2^{-126}, 2^{128}]$, the range for FP16 gets reduced to around $[2^{-14}, 2^{16}]$. Having a narrower dynamic range has proved to be detrimental, especially when representing the gradients, which typically exhibit a wider distribution compared to weights and activations. In addition, gradients have lower values that underflow (i.e. they are lower than the minimum representable number in FP16), leading to a reduced signal for the weight update as the real magnitude of the gradients is clipped to zero. A similar problem can occur if the values are greater than the maximum representable number, leading to an overflow which is typically resolved by clipping the value to that maximum. To prevent the underflow or overflow of the gradients, a popular strategy is to use loss scaling [Micikevicius et al., 2017], with its automatic variants to adapt the loss scaling during training [Noune et al., 2022, Perez, 2022].

Table 7: Comparison of floating point formats used in deep learning. E denotes the number of exponent bits, M the number of mantissa bits of a given format, and Max exp. and Min exp. are the maximum and minimum values that can be represented by the exponent, excluding special values. E5 (a) and E4(a) denote FP8 formats introduced in Noune et al. [2022], whereas E5 (b) and E4 (b) were introduced in Micikevicius et al. [2022].

| Format | E | M | Max exp. | Min exp. | Max normal | Min subnormal | bias |
|--------|---|---|----------|----------|------------|---------------|------|
| FP32 | 8 | 23 | 127 | -126 | $3.4 \times 10^{38}$ | $1.4 \times 10^{-45}$ | 127 |
| FP16 | 5 | 10 | 15 | -14 | 65504 | $6.0 \times 10^{-8}$ | 15 |
| BF16 | 8 | 7 | 127 | -126 | $3.4 \times 10^{38}$ | $9.2 \times 10^{-41}$ | 127 |
| FP8 E5 (a) | 5 | 2 | 15 | -15 | 57344 | $7.6 \times 10^{-6}$ | 16 |
| FP8 E5 (b) | 5 | 2 | 15 | -14 | 57344 | $1.5 \times 10^{-5}$ | 15 |
| FP8 E4 (a) | 4 | 3 | 7 | -7 | 240 | $9.8 \times 10^{-4}$ | 8 |
| FP8 E4 (b) | 4 | 3 | 8 | -6 | 448 | $2.0 \times 10^{-3}$ | 7 |

Due to this shorter dynamic range of FP16, deep-learning practitioners have designed the BF16 format to keep the number of exponent bits of FP32, thus maintaining a larger dynamic range. While this has proved beneficial to avoid the tuning of loss scaling, some works like Gopher [Rae et al., 2021] have shown that BF16 degrades performance in comparison with FP32, even when complementing it with techniques like stochastic rounding. This is due to the lower number of mantissa bits compared to FP16.

# B    8-bit floating point formats

The IEEE standard defines multiple floating-point formats with different bit-widths, ranging from FP256 to FP16. Proposals have recently been put forward for FP8 formats, primarily for the purpose of machine learning. The most notable ones are Noune et al. [2022] and Micikevicius et al. [2022] (additional proposals include Tesla [2021], Kuzmin et al. [2022]). The standardisation of the FP8 format is under active development by an IEEE working group, with an interim report published recently P3109 [2023].

Both of the proposals in Noune et al. [2022] and Micikevicius et al. [2022] independently recommend the use of two different FP8 formats. For gradients, the range provided by five exponent bits is required, motivating the use of an E5 format (5 exponent bits, 2 mantissa bits). For activations and weights in contrast, at least 3 mantissa bits are necessary to maintain numerical accuracy, motivating the use of an E4 format.

The E5 format of Micikevicius et al. [2022] follows the typical IEEE floating-point scheme. Due to the range limitations induced by having so few bits, for the E4 format they adjust the special value representations, assigning a single ±*NaN/inf* codeword to the bit-string containing all 1s in the exponent and mantissa, with all other bit-strings now valid values. Noune et al. [2022] adopt a similar approach, but instead use the negative zero encoding to represent the single *NaN/inf* value. These formats also differ by their default bias values. More details can be found in Table 7.

Noune et al. [2022] and Micikevicius et al. [2022] present their FP8 formats as including an additional user-defined bias value, which acts in the same way as the standard IEEE 754 bias. This is similar to multiplying the tensor by a scaling value which is a power of two. Both interpretations are used in the literature, though in practice the bias viewpoint maps better to the interface typically provided by hardware.

In this paper, if not stated otherwise, we assume the formats of Noune et al. [2022] when referring to FP8 and for experimental purposes. In practice, these formats are sufficiently similar that differences in use are likely to be minimal.

Note that Noune et al. [2022] and Micikevicius et al. [2022] assume a mixed-precision training regime, in which FP8 is used only for matrix multiplications. In other words, values are stored and accumulated in higher precision, with casting to FP8 done immediately before matrix multiplications, which themselves output in higher precision. The extent to which FP8 can be used more broadly remains an open question.

# C    The benefits of FP8 over INT8 for deep learning

The most common 8-bit format currently used in machine learning is the INT8 format, which is typically used to speed up inference . Accelerated INT8 arithmetic has been supported by previous generations of hardware, enabling users to improve the latency and throughput of their applications at inference time by quantising values (particularly weights) to INT8. However, this process can sometimes require additional tricks to maintain accuracy [Dettmers et al., 2022, Xiao et al., 2023], involving complex scaling schemes.

From a theoretical perspective FP8 has several advantages over INT8 for deep learning, which we outline below. With the introduction of hardware providing accelerated FP8 arithmetic, FP8 has the potential to supplant the use of INT8 for many inference workloads, as well as opening up the possibility of 8-bit training (generally considered infeasible for INT8).

## C.1    The distribution of values for integer versus floating-point formats

Integer formats distribute their values uniformly over the representable range, whereas floating-point formats have exponentially-spaced values. An implication of this, as described mathematically by Noune et al. [2022], is that when quantising values drawn from a unit normal distribution, floating-point formats have a high and approximately constant SNR (signal-to-noise ratio) within their dynamic range, whereas integer formats have only a narrow region with sufficiently high SNR. This phenomenon is depicted visually in the SNR plots in [Blake et al., 2023, Figure A.1.].

In practice, this means that when using floating-point formats, tensors can be scaled by an arbitrary constant factor and no change in the relative accuracy of the representation occurs (so long as values are still within the representable range of the format). This is a useful property for deep learning models where such multiplicative transformations are common. In contrast, for integer formats smaller values within the representable range become increasingly inaccurate.

## C.2 Implications for inference

For most scenarios FP8 provides as-good-as-full accuracy, with the same throughput as INT8 (assuming hardware offers the same throughput for FP8 and INT8, such as in Nvidia [2022a]) and is easier to use in practice. Our proposed FP8 schemes are also significantly simpler than the kind of methods required to attain full INT8 accuracy for the largest models [Dettmers et al., 2022, Xiao et al., 2023]. In addition, if a model has been pretrained or fine-tuned in FP8 inference, inference could be further simplified by using the scalings found at the end of training.

Despite the comparable FP8 and INT8 accuracy, there are a small number of results in the literature where the FP8 degradation is notable and cannot be neglected, such as in MobileNetV2 [Micikevicius et al., 2022, Kuzmin et al., 2022]. This may be attributable to the low numerical accuracy provided by FP8. van Baalen et al. [2023] show that when using fixed weights, integer formats can be slightly more numerically accurate than floating-point formats, in which case INT8 may help to close the gap here. However, in most applications this additional numerical accuracy isn't useful as FP8 already attains full task-accuracy, and INT8 comes at a complexity cost as outlined above.

## C.3 Implications for training

Quantisation for training is a harder problem than inference for two reasons. Firstly, the backward pass must be represented as well as the forward pass, giving a second set of tensors to be quantised that typically require a different scale. Secondly, the distribution of weight, activation and gradient tensors changes as a result of gradient updates, which is not the case for inference. There is no guarantee that the appropriate choice of formats and scalings for one point in training will be sufficient for another.

The narrow range for which integer formats have a high SNR means that integer quantisation requires careful scaling based on the distribution of values to be quantised. Given that these distributions are non-stationary during training, it is generally considered prohibitive to train with integer formats, as scalings would have to be frequently re-calculated. This non-stationarity is demonstrated by Noune et al. [2022], Kuzmin et al. [2022] in the context of FP8 training, where the approximately uniform SNR of floating-point formats is shown to enable effective FP8 training even with the scale of tensors changing over time.

The problem of having to quantise gradient tensors to FP8 is also mitigated by the fact that the FP8 E5 format used in the backward pass has a larger dynamic range than the E4 format used in the forward pass. Gradients typically use a wider spread of values than activations [Noune et al., 2022, Appendix D], making them poorly suited to integer formats, but well-represented by FP8 E5.

## D The challenges of FP8 quantisation at scale

It has been observed when training large language models that post-training quantisation becomes increasingly challenging as model scale increases, due to the presence of emergent outliers. These are defined as sequence dimensions [Bondarenko et al., 2021] or feature dimensions [Dettmers et al., 2022] in which large-magnitude values tend to be concentrated; a phenomenon which grows as model-scale increases. It has been shown that naive INT8 quantisation in the presence of these outliers substantially degrades accuracy, and various techniques have been devised to circumvent this problem, often incurring additional overheads [Bondarenko et al., 2021, Dettmers et al., 2022, Frantar et al., 2022, Xiao et al., 2023].

For the reasons outlined in Appendix C, outlier features create a particular problem for integer formats, where only a small portion of the format's numerical range has high SNR for normally-distributed values. When scaling in the presence of outliers, one can typically represent either outliers or regular

values well with integer formats, not both. Conversely, the exponential distribution of values in floating-point formats is naturally suited to representing outliers.

[Blake et al., 2023, Appendix D] model a scenario where a tensor with outlier values is quantised in both INT8 and FP8, demonstrating a substantially more accurate representation (635x higher SNR) in FP8 than in INT8. This does not preclude the possibility that sufficiently large outlier values could also cause issues for the range provided by FP8 formats, but does indicate that floating-point formats are significantly more robust than integer formats when quantising the emergent outliers that make large-scale post-training quantisation challenging.

To further mitigate the impact of outliers, recent work has shown that certain modifications can encourage models to produce fewer outliers in the first place, via the correct choice of hyperparameters [Ahmadian et al., 2023], or via changes in the attention layer [Bondarenko et al., 2023]. These methods were developed in the context of INT8 inference, but are equally applicable to FP8 training and inference in the presence of emergent outliers. The combination of these developments and the anticipated shift to FP8 inference should make quantising large models significantly easier than the community has found INT8 quantisation until now.

# E   Linear layers in GPT and Llama 2 architectures quantised to FP8

In this work we focus on quantising the linear layers of the decoder layers to FP8. The GPT and Llama decoder architectures have other types of layers including the dot-product attention that could benefit from FP8 quantisation, but we leave them for future work.

Figure 4 displays the main components of the GPT and Llama decoder layers and remarks the ones quantised to FP8. Those are:

- The attention three linear layers to project the Q, K and V matrices.
- The attention linear layer after the outputs of the heads are concatenated.
- The first feed-forward linear layer that expands the hidden dimension times four.
- The second feed-forward linear layer that contracts the hidden dimension by four.

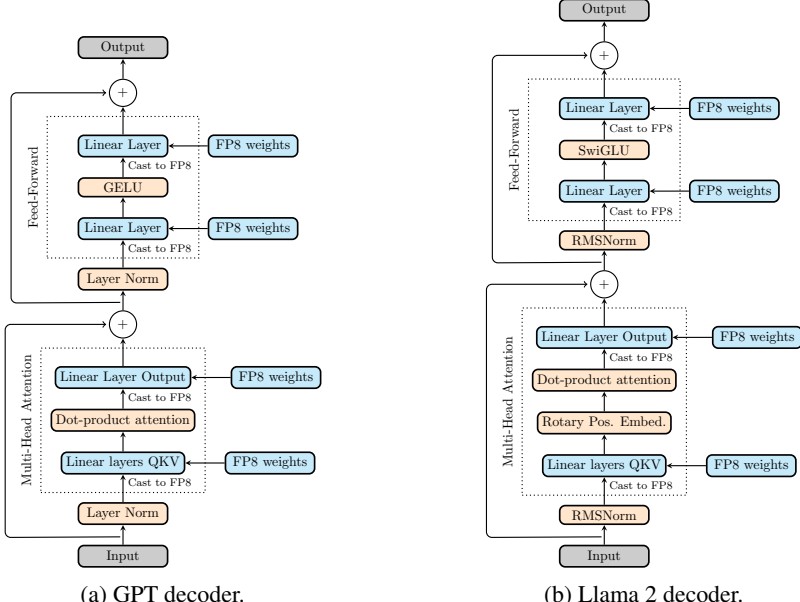

(a) GPT decoder.      (b) Llama 2 decoder.

Figure 4: Diagram remarking the linear layers quantised to FP8 (in blue) for the GPT and Llama 2 models described in Section 4.1. The rest of the layers (in orange) are kept in higher precision.

Overall, these four linear layers constitute more than 99.9% of the total compute in the decoder layer, for a typical model like the Llama 2 70B with a moderate sequence length. The FLOPs in these

four linear layers are a multiple of $d_{model}^2 \times s_l$, with $d_{model}$ being the hidden dimension and $s_l$ the sequence length, since they're based on matrix-by-vector operations. On the contrary, the dot-product attention that we keep in higher precision accounts for only a multiple of $d_{model} \times s_l^2$, since it is based on vector-by-vector operations. For hidden dimensions like the Llama 2 70B of $8192$ and moderate sequence lengths of around $4096$, the FLOPs from the dot-product attention are negligible in comparison to the FLOPs of the linear layers. However, for longer sequence lengths, attention dominates due to its quadratic cost with respect to the sequence length. The reader can find more information about FLOP counting for transformers in these two blogposts: Chen [2022], Sanger [2023].

## F    Results of the FP8-CSCALE method for inference

Figure 5 compares the FP16, FP8-AMAX and FP8-CSCALE validation accuracy for the MNLI task. FP8-CSCALE only reaches the FP16 target accuracy for a particular range of scaling biases. When the chosen scaling bias is not within that range, the validation accuracy degrades.

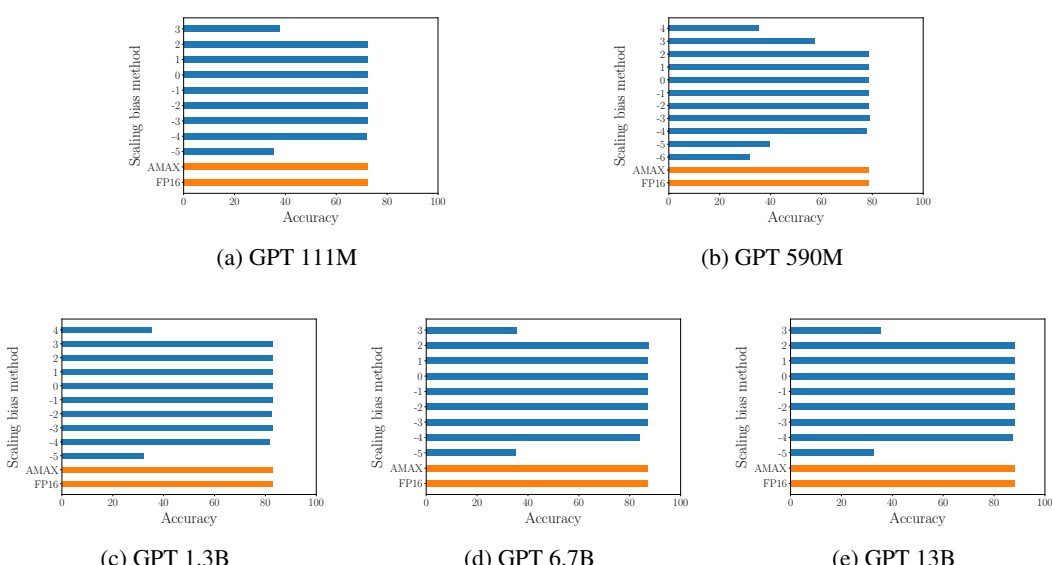

(a) GPT 111M

(b) GPT 590M

(c) GPT 1.3B            (d) GPT 6.7B            (e) GPT 13B

Figure 5: Comparison of scaling bias methods for the MNLI validation, for the different GPT sizes. Whereas the FP8-AMAX method always matches the FP16 accuracy, the FP8-CSCALE method only converges in an interval of scaling values. The specific interval that reaches at least 99.5% of the FP16 value is displayed in Table 3.

## G    Distribution of per-tensor scaling bias for inference with FP8-AMAX

In Figure 6 we display the distribution of per-tensor scaling bias for the weights and activations, for the MNLI inference setup with the GPT model detailed in Subsection 4.2. The scaling bias is computed with the FP8-AMAX method in Subsection 2.3. Whereas the scaling bias for the weights is computed only once during the PTQ from FP16 to FP8 E4, the scaling bias for the activations varies depending on the data sample. To simplify, in Figure 6 we display the statistical mode (i.e. the most frequent value) of the activation scaling bias for all the data samples in the evaluation benchmark. The four linear layers displayed (denoted as attn qkv, attn out, ff intermediate and ff output) correspond to the linear layers quantised to FP8, as explained in Appendix E. For each model size, we display the scaling bias for four decoder layer indices, with two of them being the first and the last and the two others being intermediate decoder layers.

Some observations about the plots in Figure 6 are:

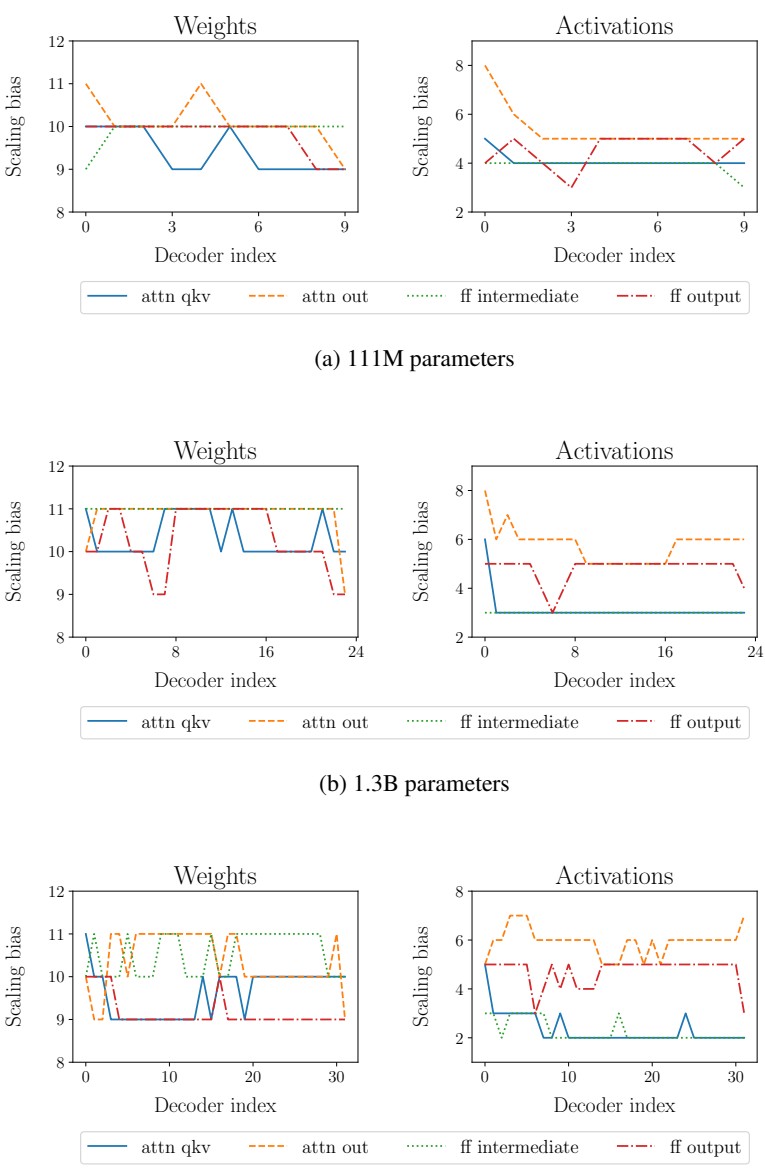

(a) 111M parameters

(b) 1.3B parameters

(c) 6.7B parameters

Figure 6: Scaling bias distribution per decoder and type of linear layer for the MNLI validation, comparing different sizes of the GPT model. The scaling bias is computed with the FP8-AMAX method in Subsection 2.3.

- Weight versus activation scaling bias: the former has greater values and a narrower distribution, spanning only three integer values versus five or six for the activation scaling bias.

- Size of the models: the scaling bias tends to vary more across decoder layer index as the model size increases.

- Type of linear layer: for the activations, the scaling biases of the attention linear layer after the outputs take greater values than the other linear layers. The scaling bias for that linear layer also varies more across decoder layer index, reaching higher values for the first decoder layers.

- The converging ranges obtained with FP8-CSCALE and displayed in Table 3 are centred around zero, whereas the scales in Figure 6 for FP8-AMAX reach greater values. This is related to the fact that FP8-AMAX chooses the maximum scaling bias per tensor that converges, but there may be other lower scaling bias that also lead to convergence. In particular, the values in Table 3 agree with the scaling biases of the activations attn qkv in Figure 6, which are the lowest ones across weights, activations and type of linear layer. A greater scaling bias value than the one shown in Table 3 with FP8-CSCALE results in overflow for that particular linear layer, which limits the maximum scaling bias that ensures convergence.

## H  Further details to run the GPT and Llama experiments

### H.1  Fine-tuning details

The fine-tuned models in Subsections 4.4 and 4.5 employ the AdamW optimiser [Loshchilov and Hutter, 2017] with (beta1, beta2) = (0.9, 0.999) and epsilon equal to 1e-5. The weight decay is 0.01 for the 111M and the 590M and 0 for the larger model sizes. Global batch size is 512 for all model sizes. We run 3 epochs for each fine-tuning task unless otherwise stated. We don't specify a gradient norm clipping. We use dropout during fine-tuning.

Further details specific to the model size are shown in Table 8. Note that we keep the learning rate constant throughout the fine-tuning since we observe that setting up a warmup plus decay didn't affect much the final validation accuracy.

Table 8: Fine-tuning hyperparameters for the fine-tuning of the GPT models in Sections 4.4 and 4.5.

| Parameters | Sequence length | Learning rate | Loss scaling |
|---|---|---|---|
| GPT 111M | 120 | 4e-5 | 512 |
| GPT 590M | 264 | 6e-5 | 4096 |
| GPT 1.3B | 528 | 3e-5 | 4096 |
| GPT 6.7B | 1040 | 8e-6 | 32768 |
| GPT 13B | 1080 | 7e-6 | 32768 |

### H.2  Hardware to run the experiments

Models were trained on IPU hardware [Graphcore, 2022a, Jia et al., 2019], using either Bow Pod$_{64}$, Bow Pod$_{16}$, IPU-POD$_{64}$ or IPU-POD$_{16}$ machines. IPU hardware allows to distribute the training and inference across multiple chips to leverage multiple levels of parallelism, which is useful to accelerate training with data replicas or to make models fit by distributing layers across multiple chips. Even though the hardware that we employ doesn't have native FP8 native, it allows for FP8 to be supported in software.

## I  Evolution of per-tensor scaling bias during training with FP8-AMAX

In Figures 7, 8 and 9 we report the evolution of the per-tensor scaling bias for the GPT model sizes of 111M, 1.3B and 6.7B parameters. The scaling bias is computed with the FP8-AMAX method in

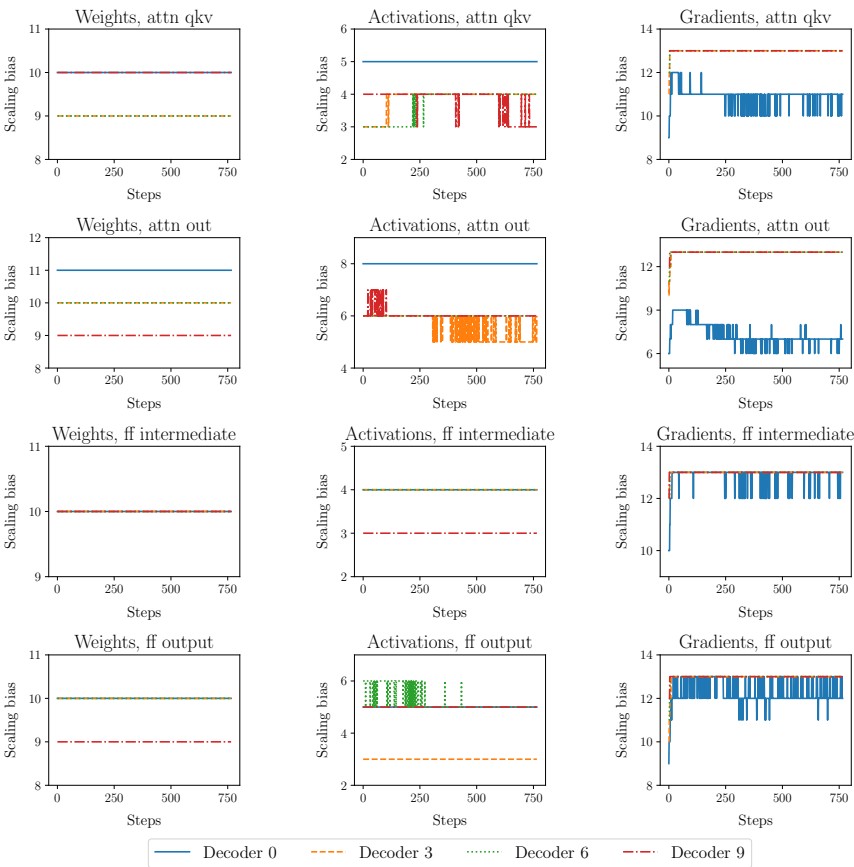

Figure 7: Evolution of the scaling bias during the fine-tuning of the 111M GPT model. The scaling bias is computed with the FP8-AMAX method in Subsection 2.3.

Subsection 2.3, and correspond to the first epoch of the fine-tuning for the MNLI task. The displayed scaling bias per training step is computed as the statistical mode (i.e. the most frequent value) of the data samples contained in the batch of that particular step. The four linear layers displayed (denoted as attn qkv, attn out, ff intermediate and ff output) correspond to the linear layers quantised to FP8, as explained in Appendix E. For each model size, we display the scaling bias for four decoder layer indices, with two of them being the first and the last and the two others being intermediate decoder layers.

Some observations from the scaling bias evolution in Figures 7, 8 and 9 are:

- Weight and activation scaling bias: both remain fairly static throughout training, with some occasional updates of just one integer value. But each decoder layer settles at a different scaling bias value, which justifies the benefit of per-tensor scales.

- Gradient scaling bias: the frequency of change is higher compared to the weight and activation scaling bias, and the values for each type of decoder layer show a sharp increase during the first training steps. This is motivated by the fact that gradients are greater at the beginning of training (i.e. need a smaller scaling bias) and lower later in the training (i.e. need a larger scaling bias).

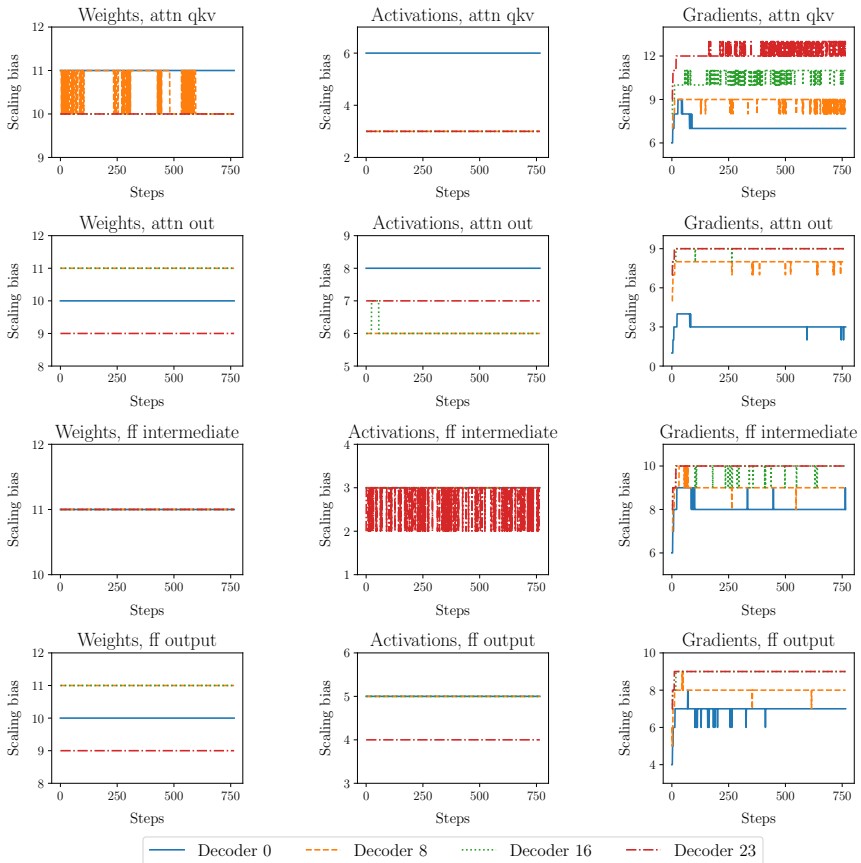

Figure 8: Evolution of the scaling bias during the fine-tuning of the 1.3B GPT model. The scaling bias is computed with the FP8-AMAX method in Subsection 2.3.

- First and last decoder layer: it has been pointed out in Noune et al. [2022] that, with constant scaling bias, some models require the first decoder layer activations and gradients to remain in FP16 for better convergence. Our plots corroborate the fact that the first layer behaves quite different to the average behaviour of the other layers. For instance, the scaling bias for the activations is typically higher for the first decoder layer index compared to the other layers, and the gradient scaling bias has instead a lower value. Concerning the last decoder layer, its gradient scaling bias takes greater values than the rest of the decoder layers and exhibit some sporadic spikes that accentuate as the model size increases (see Figure 9 for the 6.7B model as an example).

- Type of linear layer: the activation scaling bias for the attention linear layer after the outputs is higher than the rest. The gradient scaling bias change alike for the four linear layers, but for the attention linear layer after the outputs there is more difference between the first decoder layer and the rest.

- Size of the model: weight and activation scaling bias show similar plots for the three sizes, but the gradient scaling bias fluctuates much more as the size increases, spanning more integer values too.

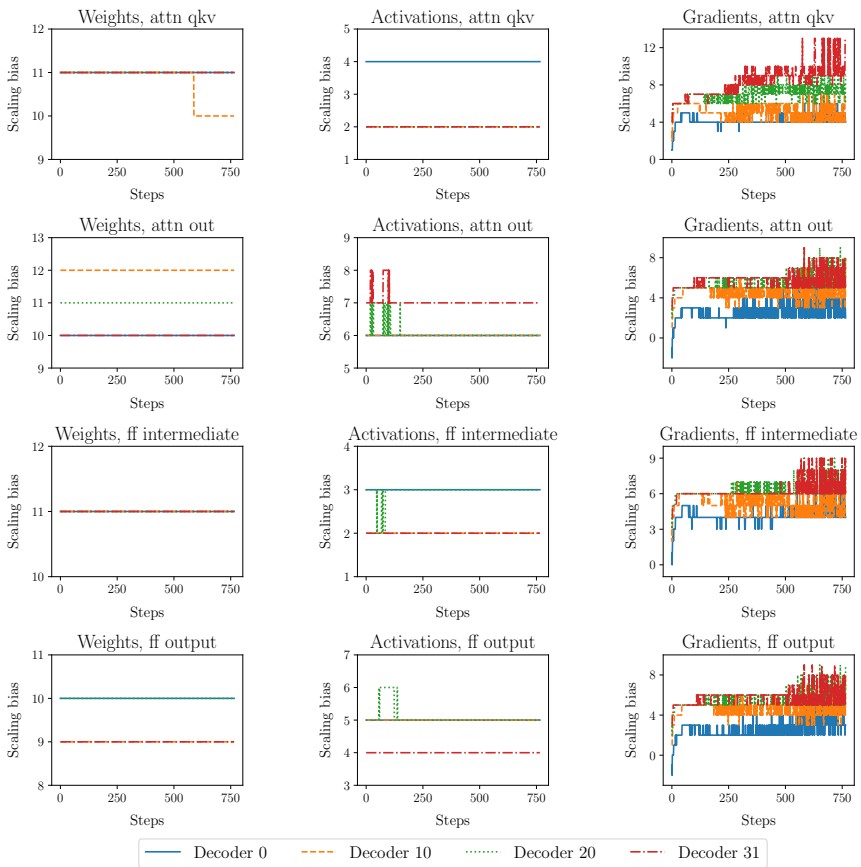

Figure 9: Evolution of the scaling bias during the fine-tuning of the 6.7B GPT model. The scaling bias is computed with the FP8-AMAX method in Subsection 2.3.

## J  Comparison of FP16 vs FP8 loss function during training

In Figure 10 we report the evolution of the loss function during the fine-tuning of the GPT model in Subsection 4.5, focusing on the MNLI task and the five model sizes. The loss functions for the other two tasks, QQP and SST2, follow a similar curve and are omitted.

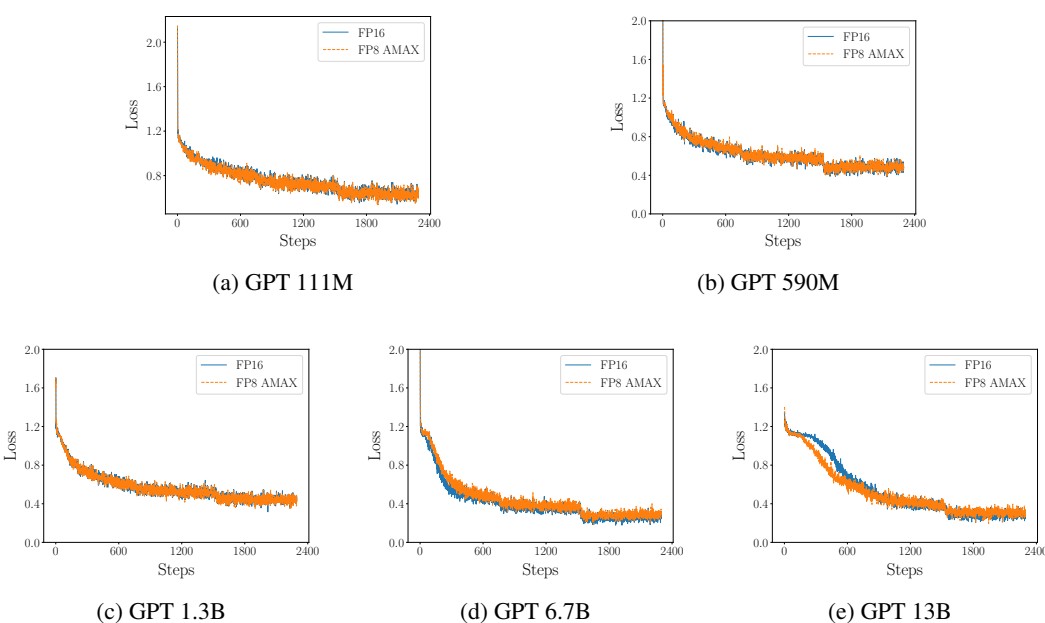

Figure 10: Fine-tuning loss for the MNLI task for the different GPT model sizes, comparing the FP16 and FP8 evolutions.

