# OpenReview forum: "Training and inference of large language models using 8-bit floating point"
_NeurIPS.cc/2023/Workshop/WANT — WANT@NeurIPS 2023 Oral_

### Official Review · Reviewer_c2p9 · 2023-10-25
**Solid work on FP8 arithmetic used for training and inference of LLMs**

**Confidence:** 4

**Review:**

Summary \
The paper addresses the problem of improving the training and inference speed of LLMs using reduced number of bits for weights and activations representation. FP8 E4 format is used for weights and activations while FP8 E5 is used for gradients. In order to prevent the underflow or overflow when casting from FP16 to FP8 E4 the scaling bias schemes are introduced. Either fine-tuned or non-finetuned models with linear layers in FP8 format show small or no performance drop comparing to the ones in FP16 format.

Quality: the proposed method is evaluated on two popular family of models, GPT and Llama 2 for three GLUE tasks and shows its efficiency.\
Clarity: the paper is well-written, well-organized and easy-to-follow.\
Originality: the proposed schemes are not completely original being a minor improvement in terms of scientific novelty.\
Significance: compression and acceleration of LLMs are of great interest in industry.

Pros:
* Small or no drop in performance according to evaluations stated.
* Extensive analysis, possible limitations of FP8-AMAX and FP8-CSCALE methods are discussed.
* Promising results for industry and academia.

Cons:
* Lack of scientific novelty.
* Too much material is moved to Supplementary (not critical).

---

### Official Review · Reviewer_bJPH · 2023-10-25
**An important step to make training and inference of LLMs in FP8 format feasible**

**Confidence:** 4

**Review:**

Authors research the problem of doing training and inference of large LLMs in FP8 format, focusing specifically on linear layers. The paper is well written and provides thorough background of problems arising from doing computations in FP8. They propose to solve them by carefully selecting scaling biases and explore two different strategies for doing it. A series of extensive computational experiments on modern LLMs proves the value of the propose approach.

This paper is important for establishing the foundation for training LLMs in FP8. It would be interesting to see if the ideas discussed in the paper can be broadcasted to other layers of LLMs, especially attention ones. Given the arbitrary sequence length and potentially higher variance in activation values arising from it.

Also, in my opinion, the title of the paper is too bold. I believe it should mention that the paper is focused on linear layers. Right now the title reads like the paper provides a recipe for training complete LLMs in FP8. However, this is not the case.

---

### Meta-Review · Area_Chair_b9uQ · 2023-10-27

**Recommendation:** Accept (Poster)
**Confidence:** 4

**Metareview:**

Both reviewers acknowledge the practicality of the proposed approach and in-depth study, raising concerns regarding the limited technical novelty. Moreover, Reviewer bJPH points out that the title of the paper slightly overclaims the contributions of the method. Overall, despite some shortcomings, the paper is of interest to community especially as the support for FP8 hardware gains traction.

---

### Decision · Program_Chairs · 2023-10-28

**Decision:**

Accept (Oral)

**Comment:**

We thank the authors for their time and contribution to WANT and we are pleased to share that after the reviewing process the paper has been accepted. Congratulations! We encourage the authors to consider reviewers' feedback for the improvement of the camera-ready version. We hope to see you in person at the workshop and brainstorm on efficient training research together!